# Temporal modulation of collective cell behavior controls vascular network topology

Esther Kur[1], Jiha Kim[1], Aleksandra Tata[1], Cesar H Comin[2], Kyle I Harrington[3], Luciano da F Costa[2], Katie Bentley[3]*, Chenghua Gu[1]*

[1]Department of Neurobiology, Harvard Medical School, Boston, United States; [2]Instituto de Física de São Carlos, University of Sao Paulo, Sao Carlos, Brazil; [3]Center for Vascular Biology Research, Department of Pathology, Beth Israel Deaconess Medical Center, Harvard Medical School, Boston, United States

**Abstract** Vascular network density determines the amount of oxygen and nutrients delivered to host tissues, but how the vast diversity of densities is generated is unknown. Reiterations of endothelial-tip-cell selection, sprout extension and anastomosis are the basis for vascular network generation, a process governed by the VEGF/Notch feedback loop. Here, we find that *temporal* regulation of this feedback loop, a previously unexplored dimension, is the key mechanism to determine vascular density. Iterating between *computational* modeling and in vivo live imaging, we demonstrate that the rate of tip-cell selection determines the length of linear sprout extension at the expense of branching, dictating network density. We provide the first example of a host tissue-derived signal (Semaphorin3E-Plexin-D1) that accelerates tip cell selection rate, yielding a dense network. We propose that *temporal* regulation of this critical, iterative aspect of network formation could be a general mechanism, and additional temporal regulators may exist to sculpt vascular topology.

*For correspondence: kbentley@ bidmc.harvard.edu (KB); Chenghua_Gu@hms.harvard.edu (CG)

**Competing interests:** The authors declare that no competing interests exist.

## Introduction

The unique vascular topology of different organs exists for organ-specific functions. Different vascular network densities determine the specific amount of oxygen and nutrients to be delivered to each host tissue. Development of new vascular networks depends upon two types of specialized endothelial cells that work together: (1) The endothelial 'tip cell', which is located at the front of a growing vessel and guides its extension by sensing and responding to environmental cues, analogous to the axonal growth cone (*Gerhardt et al., 2003*; *Kurz et al., 1996*). (2) 'Stalk cells', which trail behind the tip cell and elongate the sprout. Tip and stalk cell identities are primarily controlled by the Dll4-Notch lateral inhibition pathway, which is activated in endothelial cells in response to VEGF from the local environment. VEGF-induced Dll4 activates Notch1 on the neighboring cell, leading to the down-regulation of VEGF receptor levels (*Figure 1B*) (*Hellström et al., 2007*; *Leslie et al., 2007*; *Lobov et al., 2007*; *Suchting et al., 2007*),(*Benedito et al., 2009*; *Phng et al., 2009*; *Roca and Adams, 2007*). Thus, lateral inhibition between endothelial cells generates an alternating pattern of active tip cells (Dll4 high) and inhibited stalk cells (Dll4 low). Moreover, tip cell selection is a dynamic process, and tip cell identity is transient (*Arima et al., 2011*; *Jakobsson et al., 2010*). Reiteration of tip cell selection, sprout extension, and connection of neighboring sprouts (anastomosis) is the basis for building a sophisticated vascular network (*Adams and Eichmann, 2010*; *Carmeliet, 2000*). Although this VEGF/Notch signaling pathway has been well studied and found to be conserved

**eLife digest** Many animals have a network of blood vessels that supplies oxygen and nutrients to every part of the body. Each organ contains a unique pattern of blood vessels; some have lots of densely packed vessels, while others have fewer vessels that are more widely spaced.

New blood vessels typically form by sprouting from the side of pre-existing vessels. This involves the endothelial cells that line the inner wall of blood vessels moving outwards to create a sprout that is made up of 'tip cells' and 'stalk cells'. Tip cells are found at the front of the growing vessels and encourage the formation of new sprouts, while the stalk cells trail behind and elongate the sprout.

Two signaling pathways that involve two proteins called VEGF and Notch interact with each other to control which cells become tip cells and which become stalk cells. Cells with higher levels of VEGF signaling will become tip cells. These cells also activate Notch signaling, which in turn blocks VEGF signaling in their neighboring cells. This feedback mechanism enables a new sprout to form by forcing cells present around a newly formed tip cell to become stalk cells. However, it was still not understood how the different organs develop blood vessel networks with different densities.

In 2011, researchers revealed that two other proteins, Semaphorin3E and its receptor Plexin-D1, are expressed in tip cells in the back of the eye in mice and control the VEGF/Notch signaling pathway. Now Kur et al. – including some of the researchers involved in the 2011 work – have used a combination of predictive computer simulations and experimental approaches to understand this interaction in more detail.

The analysis showed that Semaphorin3E and Plexin-D1 speed up VEGF/Notch signaling, which causes new tip cells to form at a faster rate, and results in a more densely packed network of blood vessels. For example, in mice that lack Semaphorin3E and Plexin-D1, VEGF/Notch signaling was slower and new tip cells formed more slowly, which resulted in the blood vessel network at the back of the mice's eyes being less dense.

Kur et al. propose that different organs have different 'molecular metronomes' that control the pace of VEGF/Notch signaling. A fast acting metronome would yield a dense network, while a slower one would form a less dense network. This helps to explain how diverse densities of blood vessel networks are formed in different organs. This work may aid efforts to develop therapeutic approaches for controlling the development of new blood vessels in cancers and other diseases.

among different vascular beds and species, how this central pattern generator is modified by various target tissue-specific signals to yield diverse network topologies is not known.

Here, we propose a general principle of how collective cell behavior determines the diverse densities of different networks: the generation of vascular topologies depends heavily on the *temporal* regulation of tip cell selection. Integrated simulations predict that as cell neighborhoods change, due to anastomosis or cell rearrangement events, lateral inhibition patterns will necessarily be disrupted, requiring continual re-selection of new tip cells (*Bentley et al., 2009*; *2014a*). In fact, mouse genetics experiments demonstrated that tip cell numbers are positively correlated with the branching points of the network (*Hellström et al., 2007*; *Kim et al., 2011*). Therefore, the *length of time* it takes to establish (and re-establish) the alternating pattern of tip and stalk cells may be a missing, critical determinant of vascular topology (*Bentley et al., 2014b*; *2014c*). Here, we took an integrated approach combining computational modeling, mouse genetics, and in vivo endothelial cell tracking to determine whether tip/stalk patterning can be temporally modulated to generate different topologies. We hypothesize that the frequency of tip cell selection determines the length of linear extension vs. branching, thus dictating the density of the network.

To begin to test this hypothesis, it is crucial to analyze dynamic single cell behavior and collective movement in the context of network formation (*Arima et al., 2011*; *Jakobsson et al., 2010*). Previously, we used static analyses of the postnatal mouse retina as a model to understand how neural signals shape vascular topology (*Kim et al., 2011*). We discovered that retina ganglion cell-derived Semaphorin3E (Sema3E) and its receptor Plexin-D1, which is expressed in endothelial cells at the front of actively sprouting blood vessels, control the VEGF/Notch pathway via a feedback mechanism. Mice lacking either Sema3E or Plexin-D1 exhibited an uneven vascular growth front and a

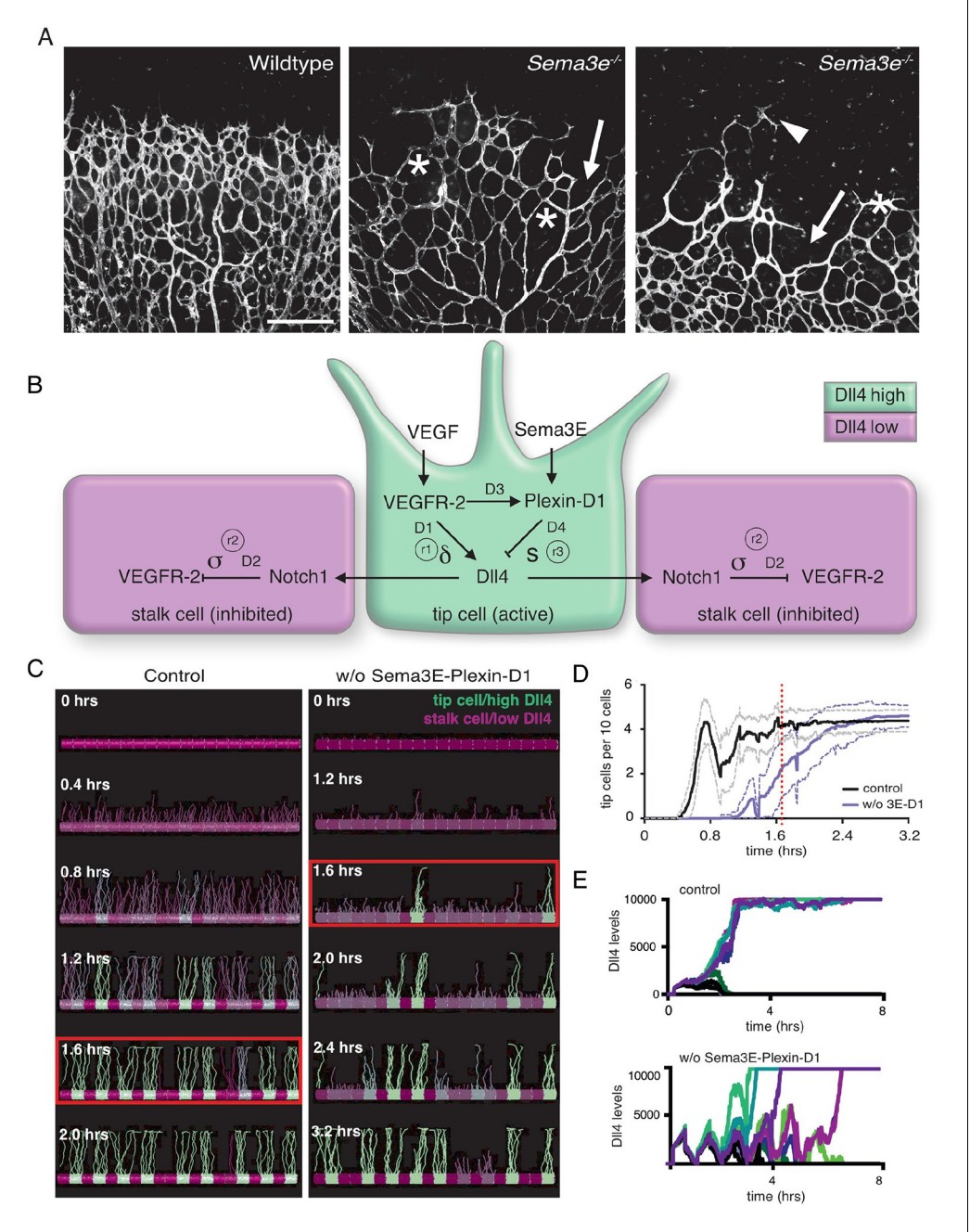

**Figure 1.** Calibrated computational model predicts delayed tip cell selection in the absence of Sema3E-Plexin-D1 signaling. (A) Whole-mount vascular staining (Isolectin B4) of retinas from *Sema3e*[-/-] and wildtype littermates at P4. The mutant vasculature exhibits a reduced number of tip cells and branching points (asterisks) and an uneven growth front (arrows and arrowheads). Scale bar: 500 μm. (B) Feedback between the VEGF/Notch and Sema3E-Plexin-D1 signaling pathways included in the extended agent-based computational model of tip cell selection. D1-D4: transcriptional delays. r1-r3: recovery delays representing degradation. δ, s, σ: change in expression levels in response to receptor activation. (C) Simulated tip cell selection. Colors represent Dll4 levels on a continuum from purple (low) to green (high). The red boxes highlight a time frame in which a salt and pepper pattern has formed in the control vessel, while in the absence of Sema3E-Plexin-D1 signaling, only few early tip cells have been selected. (D) Average number of selected tip cells in simulated vessels. At a timepoint where the simulated control vessel (black line) already exhibits an alternating pattern of tip and stalk cells, the simulated vessel lacking Sema3E-Plexin-D1 signaling (blue line, for a given set of parameter values: δ =5, s=3) shows a 50% reduction in tip cells. Thin lines: standard deviation. n=50. (E) In silico Dll4 levels in single endothelial cells during simulated tip cell

*Figure 1 continued on next page*

*Figure 1 continued*

selection. In the control situation (top), Dll4 levels quickly stabilize. In the absence of Sema3E-Plexin-D1 signaling (bottom) Dll4 levels fluctuate in near synchrony before they finally stabilize.

reduction of tip cells that resulted in a less branched network compared to their wildtype littermate controls (*Kim et al., 2011*) (*Figure 1A*). However, it is not clear how this phenotype is generated: specifically, how the Sema3E-Plexin-D1 feedback mechanism regulates VEGF/Notch signaling at a dynamic cellular level, and whether changes in temporal modulation of this pathway lead to the overall vascular topology phenotype.

To begin to understand how Sema3E-Plexin-D1 signaling modifies vascular topology formation in a dynamic, spatiotemporal manner, we took advantage of an existing agent-based computational model (the 'MemAgent-Spring Model' or MSM) that simulates the cellular processes during tip cell selection making explicit the time it takes for gene expression (e.g. transcription/translation) changes to occur (*Figure 1B,C* – note time delay parameters D1 and D2) (*Bentley et al., 2008*; *2009*). The MSM has been tested against numerous independent experimental data sets and validated as predictive of new mechanisms in vivo/in vitro (*Bentley et al., 2014a*; *Guarani et al., 2011*; *Jakobsson et al., 2010*). To now simulate tip cell selection in the context of Sema3E-Plexin-D1 crosstalk signaling with VEGF/Notch signaling (*Fukushima et al., 2011*; *Kim et al., 2011*) the MSM model was extended by adding four new parameters (*Figure 1B*, *Video 1–5*), with sensitivity analyses and calibration simulations performed, which include modulation of the existing parameter (δ) representing the induction level of Dll4 by VEGFR-2 activation (See methods section). These four new parameters represent the time delay for induction of Plexin-D1 by VEGF (D3), the time delay (D4) and strength (s) of the reduction of Dll4 levels in response to Sema3E-Plexin-D1 signaling, based on the experimental data previously shown (*Kim et al., 2011*), as well as the degradation rate of Plexin-D1 (r3). Loss of Sema3E-PlexinD1 signaling was simulated by setting all Plexin-D1 levels to zero. Loss of function simulations recapitulated the two prominent features of *Plxnd1*[-/-] and *Sema3e*[-/-] mutant retinal vasculature remarkably

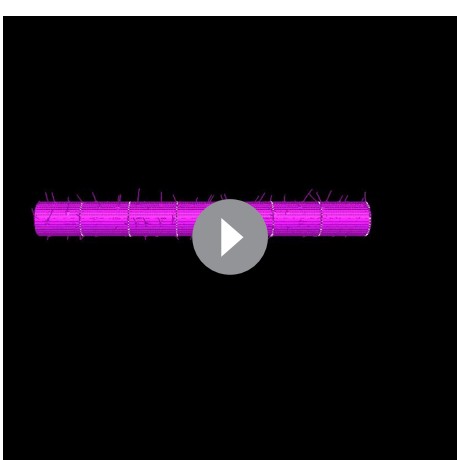

**Video 1.** Calibration of time delay parameters (induction of Plexin-D1, reduction of Dll4 levels). Simulation of tip cell selection with Sema3E-Plexin-D1 related regulatory time delays equal to the time it takes for pVEGFR-2 to up-regulate Dll4 (D3+D4=D1). Color indicates cell Dll4 levels (green = high, purple = low).

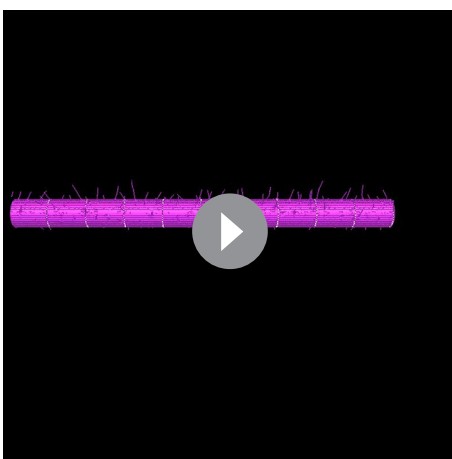

**Video 2.** Calibration of time delay parameters (induction of Plexin-D1, reduction of Dll4 levels). Simulation of tip cell selection with Sema3E-Plexin-D1 related regulatory time delays slightly slower than the time it takes for pVEGFR-2 to up-regulate Dll4 (D3 +D4=D1+1). Dll4 levels flash irregularly between very high and very low with no stable selection possible. Color indicates cell Dll4 levels (green = high, purple = low).

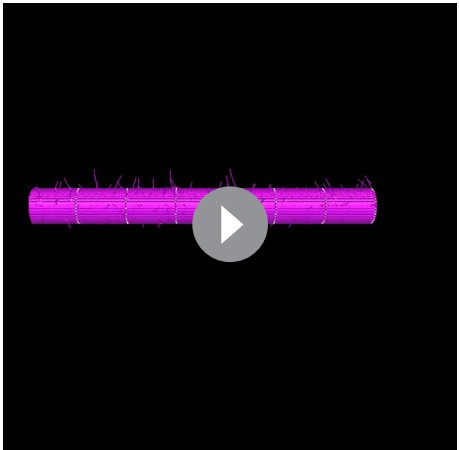

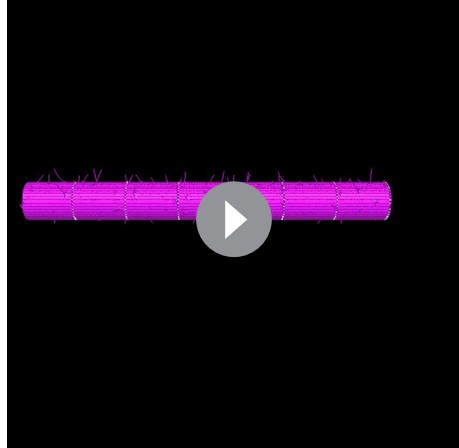

**Video 3.** Calibration of time delay parameters (induction of Plexin-D1, reduction of Dll4 levels). Simulation of tip cell selection with Sema3E-Plexin-D1 related regulatory time delays slightly faster than the time it takes for pVEGFR-2 to up-regulate Dll4 (D3 +D4=D1-1). Dll4 levels flash irregularly between very high and very low with no stable selection possible. Color indicates cell Dll4 levels (green = high, purple = low).

**Video 4.** Calibration of Plexin-D1 degradation rate. Simulation of tip cell selection with slowed degradation of Plexin-D1 allowing it to affect transcription of Dll4 for one timestep (12 s) longer. Dll4 levels flash irregularly between very high and very low with no stable selection possible. Color indicates cell Dll4 levels (green = high, purple = low).

closely: In vivo, mutant vascular networks exhibit fewer tip cells and 1.5–2 fold reduction in branching points, as well as an uneven growth front (compare *Figure 1A* with 1C [red boxes]) (*Kim et al., 2011*). Furthermore, the dynamic nature of the simulations provided a novel insight to explain how this phenotype is generated. Simulations predict that higher Dll4 levels in the cells generates a shift towards synchronized Dll4 fluctuations overtime in contiguous cells, as they collectively battle more strongly via lateral inhibition negative feedback, causing an overall delay in amplification of differences needed to select the alternating pattern of tip

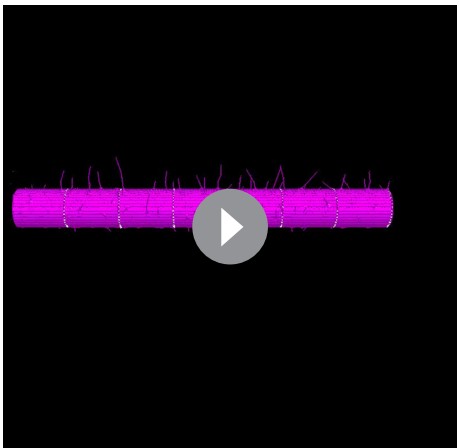

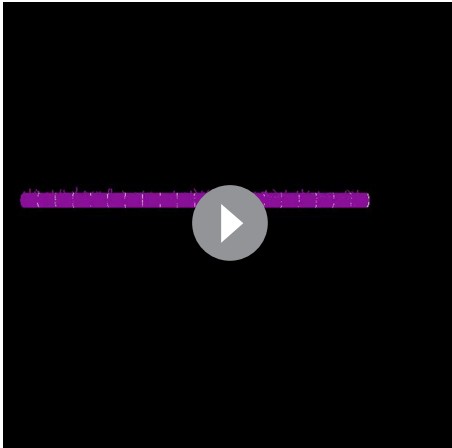

**Video 5.** Calibration of Plexin-D1 degradation rate. Simulation of tip cell selection with slowed degradation of Plexin-D1 allowing it to affect transcription of Dll4 for two timesteps (30 s) longer. Dll4 levels flash irregularly between very high and very low with no stable selection possible. Color indicates cell Dll4 levels (green = high, purple = low).

**Video 6.** Simulation of a vessel with 20 endothelial cells in the absence of Sema3E-Plexin-D1 signaling. Extended regions occur with no sprouting as cells battle for longer undergoing fluctuations as the Dll4 up-regulation is higher. These eventually resolve and tip cells are selected across the whole region.

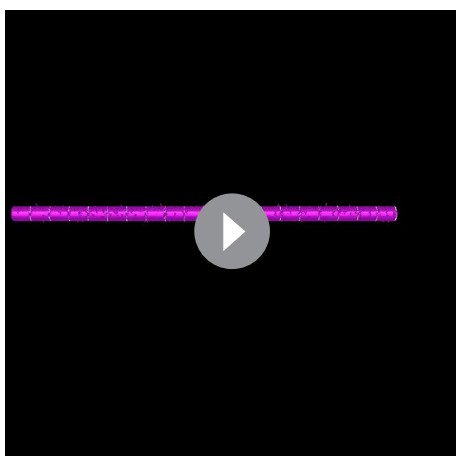

**Video 7.** Simulation of a wildtype vessel for comparison with *Video 6*. The selection occurs much faster and more regularly than in the vessel without Sema3E-Plexin-D1 signaling.

and stalk cells (*Figure 1C–E*, *Video 6,7*). Therefore, computational simulations suggest that Sema3E-Plexin-D1 signaling enhances the speed and frequency of tip cell selection and thus increases the density of retinal vascular networks.

We next examined single cell behavior in a mosaic environment, because lateral inhibition requires collective coordination, and the relative, not absolute, levels of Dll4 expression among neighboring cells determine the outcome of the tip cell selection process (*Jakobsson et al., 2010*). A *Plxnd1*$^{-/-}$ cell will behave differently in competition with a second *Plxnd1*$^{-/-}$ cell than in competition with a wildtype cell. Thus, we investigated whether the cell autonomous function of Sema3E-Plexin-D1 signaling in the direct competition process between two neighboring endothelial cells can drive changes to the tip cell selection dynamics. In a simulated mosaic vessel, where cells lacking Sema3E-Plexin-D1 signaling are intermingled with normal cells, the contribution of cells lacking Sema3E-Plexin-D1 signaling to the tip cell population was predicted to increase to 68–74% (robustly across a range of parameter values: δ =4–6, s=2–4) in comparison to control cells (*Figure 2A,B*, *Video 8*), with an increased speed of patterning compared to a vessel entirely lacking Sema3E-Plexin-D1 signaling. This result suggests that normally, Plexin-D1 cell-autonomously suppresses the tip cell phenotype.

To test the direct competition of *Plxnd1*$^{-/-}$ cells and wildtype cells in vivo, we performed mosaic analysis using mice with tamoxifen inducible loss of *Plxnd1* expression in the vasculature (*Cdh5-Cre-ER*$^{T2}$*; Plxnd1*$^{flox/flox}$). In the sprouting front of the wildtype retina, cells expressing Plexin-D1 were equally distributed at both tip cell and the adjacent stalk cell positions (*Figure 2C*, left panel, *Figure 2D*), showing they have no preference for either position. However, at approximately 45% of *Plxnd1*$^{-/-}$ mosaicism, 75% of the *Plxnd1*$^{-/-}$ cells became tip cells (*Figure 2C*, middle panel, *Figure 2D*) indicating that these cells do have a competitive advantage over Plexin-D1 expressing wildtype cells during tip cell selection. In contrast, in control experiments using mice with tamoxifen inducible expression of GFP (*Cdh5-Cre-ER*$^{T2}$*; Z/EG*$^{+}$), GFP positive cells showed no preference for either position (*Figure 2C*, right panel, *Figure 2D*), demonstrating that preferential tip cell occupancy of mutant cells is due to lack of Sema3E-Plexin-D1 signaling. Taken together, in silico prediction and in vivo mosaic retinal analyses demonstrate that Sema3E-Plexin-D1 signaling suppresses the tip cell phenotype in a cell-autonomous manner. Interestingly, this cell-autonomous effect at single cell level contrasts with the effect of Sema3E-Plexin-D1 signaling at the collective level, where more new tip cells are selected in the presence than in the absence of Sema3E-Plexin-D1 signaling (*Figure 1*). This finding highlights the differences between collective and cell autonomous behaviors, and the challenge of intuiting one from the other.

Having confirmed that the calibrated set of parameters is valid to model contributions of the Sema3E-Plexin-D1 pathway to VEGF/Notch patterning and to predict new in vivo data of a static nature, we next employed the model to predict the effect of Sema3E-Plexin-D1 signaling on the rate of tip cell selection during the fuller dynamic network formation processes in which tip and stalk cell identities are constantly re-defined during cell rearrangement/position switching. Here, we used a new 'MSM-CPM' model that has been previously extended, parameterized and experimentally validated to simulate tip/stalk patterning together with cellular rearrangements within a vascular sprout (*Bentley et al., 2014a*). In this model, a cell can move within the adhered collective of the sprout powered by multiple local junctional movements. By incorporating the newly calibrated Sema3E-PlexinD1 signaling extension the MSM-CPM model we simulated cell rearrangements in vascular sprouts in the presence or absence of Sema3E-Plexin-D1 signaling (*Figure 3A,B*, *Video 9,10*). Simulations predicted that in the presence of Sema3E-Plexin-D1 signaling, the tip cell

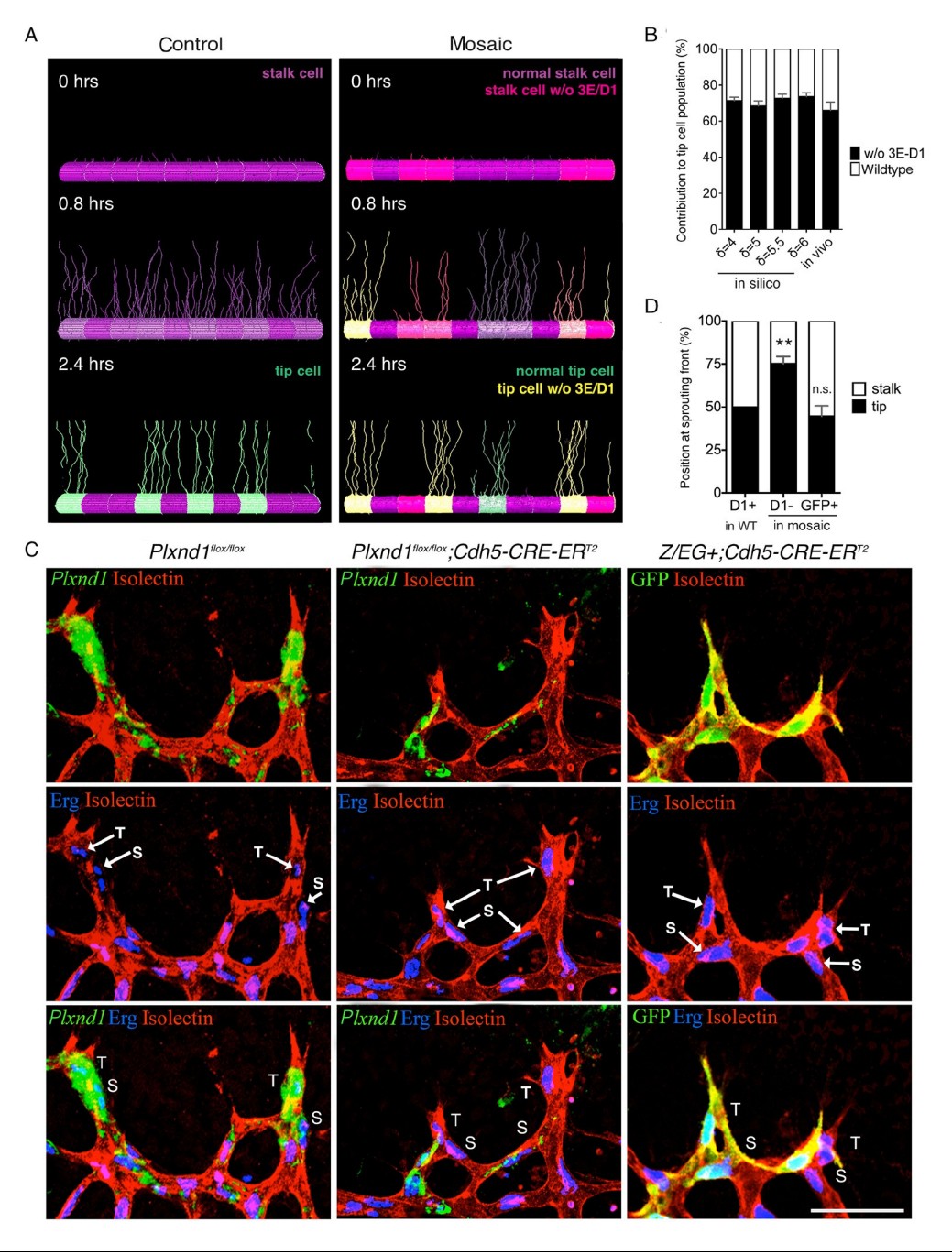

**Figure 2.** Sema3E-PlexinD1 signaling is cell-autonomously required to suppress tip cell identity. (**A**) Simulated tip cell selection in a mosaic vessel with 50% mutant cells placed randomly. Cells without Sema3E-Plexin-D1 signaling are indicated by bright pink color, which turns to yellow if Dll4 levels increase. For wildtype cells: purple = low Dll4, pink = high Dll4. (**B**) Comparison of simulated and in vivo contribution to the tip cell population by cells lacking Sema3E-Plexin-D1 signaling at 45% mosaicism. A range of δ values simulates different strengths of loss of Sema3E-Plexin-D1 signaling. Simulations, n=50, in vivo, n=6. Data is represented as mean +/- SEM. (**C**) Analysis of the occupation of tip or stalk cell position by *Plxnd1* expressing wildtype cells in control retina (left), and *Plxnd1*[-/-] cells (middle) or GFP+ cells (right) in mosaic retinas at P5. Red: vascular membrane staining (Isolectin B4), blue: vascular nuclear staining (α-ERG), green: in situhybridization (left, middle), α-GFP staining (right). (**D**) Quantification of c. n.s.= not significant, **p=0.0033. WT retinas, n=6; *Plxnd1*[-/-] mosaic retinas, n=6; GFP+ mosaic retinas, n=4. Scale bar: 50 μm. Data is represented as mean +/- SEM.

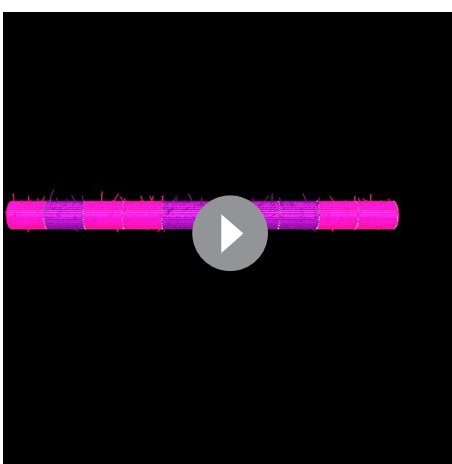

**Video 8.** Simulated mosaic vessel with 50% mutant cells placed randomly. Cells without Sema3E-Plexin-D1 signaling are indicated by bright pink color, which turns to yellow if Dll4 levels increase. For wildtype cells: purple = low Dll4, pink = high Dll4 as before. This chimera achieves a 75% contribution of mutant cells to the tip.

is repeatedly overtaken by another cell. In contrast, in the absence of Sema3E-Plexin-D1 signaling, tip cell overtaking frequency is reduced by factor of 1.26; a given tip cell occupies the tip cell position for a longer time (*Figure 3C*). Given the MSM is a qualitative not quantitative model we also performed simulations over a range of parameters and found that if the strength of Dll4 upregulation by VEGF (δ) is increased then this delay in overtaking is further exaggerated; a given tip cell occupies the tip cell position longer (*Figure 3C*).

To test experimentally if tip cell selection and overtaking rates are slowed down in the absence of Sema3E-Plexin-D1 signaling, we next performed live endothelial cell tracking to follow the behavior of individual cells in an actively forming vascular network. Previously, live imaging with single cell resolution has mainly been described in sprouting assays using aortic rings (*Arima et al., 2011*) or embryoid bodies (EBs) generated from ES cells (*Jakobsson et al., 2010*). However, those assays primarily give rise to simple linear sprouts with less frequent branching than observed during retinal angiogenesis. To better mimic in vivo temporal and spatial events within a highly branched network, we developed a new ex vivo system, using explants from embryonic lungs. Whole embryonic lungs (E12.0) were embedded within a collagen matrix in a tissue culture dish containing medium supplemented with recombinant human VEGF. After one day, endothelial sprouts grow out of the explant (*Figure 4—figure supplement 1A*) with tip cells extending filopodia into the collagen matrix (*Figure 4—figure supplement 1B*). We observed branching events at various positions of the sprouts (*Figure 4B*), as well as anastomosis (*Figure 4—figure supplement 1C*), showing that all steps of the in vivo angiogenic sprouting process are recapitulated in our system even though there is no gradient of VEGF signaling. Lung explants endogenously express *Sema3e* and *Plxnd1* (*Figure 4—figure supplement 1D*). We could also detect Plexin-D1 and Dll4 in sprouting endothelial cells (*Figure 4—figure supplement 1E,F*). We next performed long-term live imaging of wildtype and *Plxnd1*[-/-] explants and analyzed tip cell selection frequency. In contrast to the computational model where a tip cell selection event can only occur via a switch between two cells at the tip of the sprout (*Figure 3A*), in the live imaging assay, tip cell selection events occur in the following distinct categories (*Figure 4A,B*): a positional switch between a tip and a stalk cell in a linear sprout (switch), the selection of a new tip cell at the front of the sprout (branch, type I) or at more proximal sites (branch, type II). Single cells were tracked manually using a nuclear live stain. Consistent with the computational model prediction, in vivo live imaging data indeed showed a significant delay in the selection of new tip cells in *Plxnd1*[-/-] vascular sprouts compared to wildtype sprouts. The overall appearance of new tip cells, i.e. the tip cell selection frequency, was slowed down by factor 1.5 in the mutants (*Figure 4C,D*, *Video 11,12*). When analyzing the different categories separately, the number of events per total imaging time was significantly reduced in the categories 'switch' and 'branch, type II' in the absence of Sema3E-Plexin-D1 signaling (*Figure 4E*).

Finally, to directly test experimentally whether the reduced tip cell selection rate observed by live imaging in our *Plxnd1*[-/-] lung explant indeed leads to a less branched network over time, we analyzed the topology of the lung explant under the same culture conditions as the live imaging paradigm. Using a computational method (*Figure 4—figure supplement 2* and detailed description in the material and methods section) to analyze the number of branching points in an unbiased way we found a significant reduction in branching points of the network (*Figure 4F,G*). These data further demonstrate that a slowed tip cell selection rate results in a less branched vascular network.

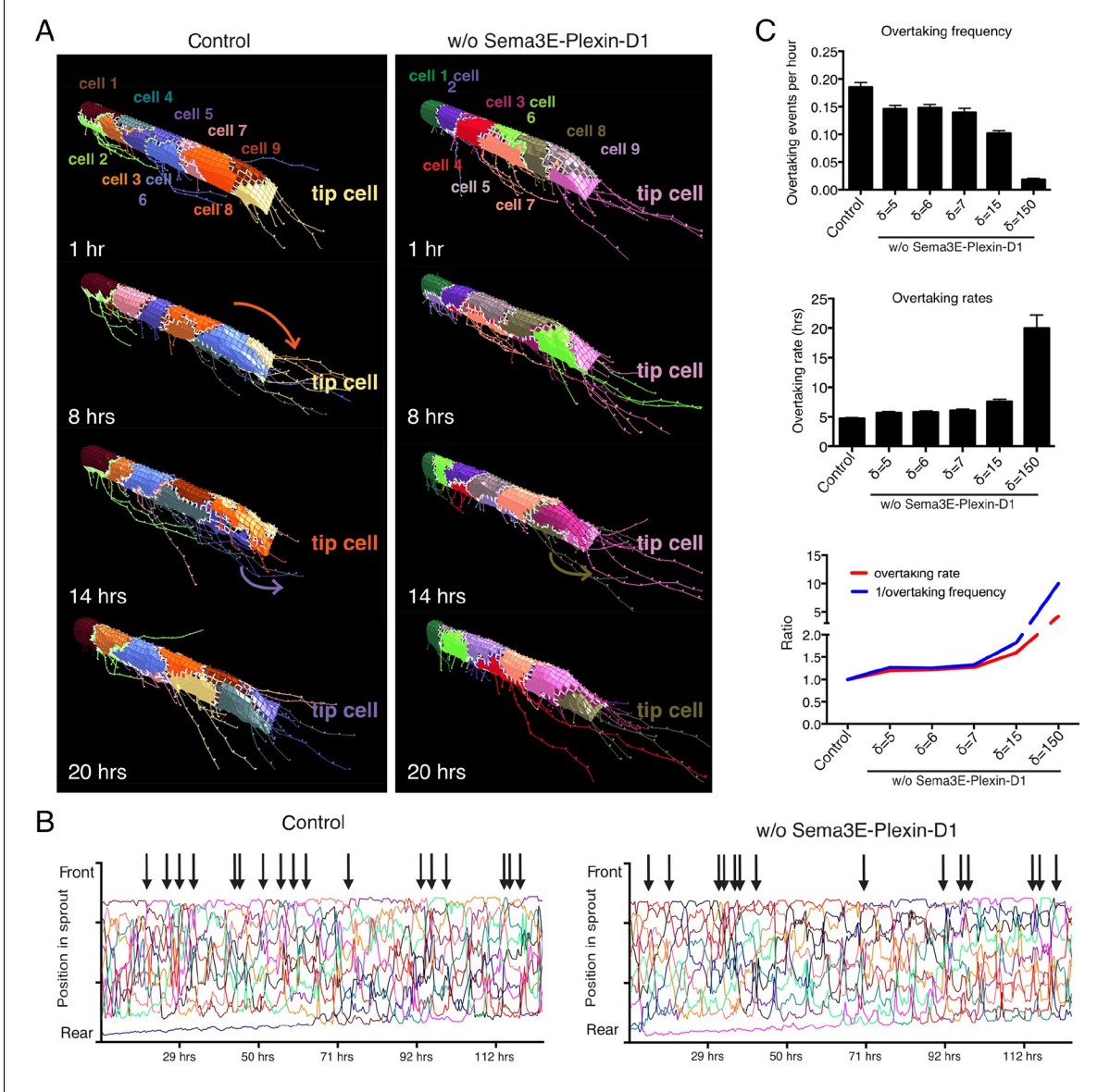

**Figure 3.** Computational simulation predicts that lack of Sema3E-Plexin-D1 signaling leads to prolonged tip cell occupancy and reduced tip cell overtaking frequency. (**A**) Single frames of cell rearrangements in simulated sprouts of 10 cells. VEGF gradient extends in direction of vessel. (**B**) Kymograph plots of cell rearrangements in simulated sprouts. Each line represents one endothelial cell. Arrows indicate overtaking events at the tip cell position in a and b. (**C**) Quantification of overtaking events at the tip cell position in the presence and absence of Sema3E-Plexin-D1 signaling. A range of δ values simulates different strengths of loss of Sema3E-Plexin-D1 signaling. The setting δ=5, which matched loss of Sema3E-Plexin-D1 signaling in other conditions in the paper exhibits a 1.26 slower tip cell overtaking frequency (events/hour.) As δ increases there is a clear trend towards slower tip cell overtaking across δ values. Data is presented as mean +/- SD, n=50.

Together, the ex vivo and in silico results demonstrate that Sema3E-Plexin-D1 signaling modulate the pace of tip cell selection. In the absence of Sema3E-Plexin-D1 signaling, the rate of tip cell selection is reduced, which overall leads to longer linear sprout extension with less frequent branching substantially influencing the architecture of the growing vasculature and resulting in a less dense network, as seen in the *Plxnd1* and *Sema3e* mutant lung explant as well as in retina (*Kim et al., 2011*).

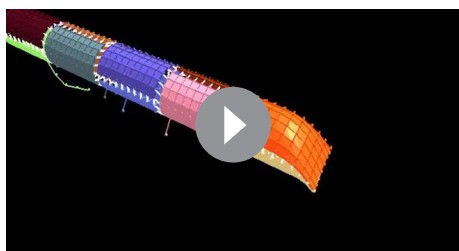

**Video 9.** Simulation of normal cell rearrangement and tip cell overtaking in a sprout consisting of ten cells, two per vessel cross section. VEGF gradient extends in the direction of the sprout. Each cell indicated by a different color.

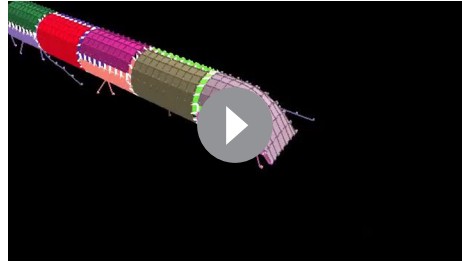

**Video 10.** Simulation of cell rearrangement in a sprout, in the absence of Sema3E-Plexin-D1, consisting of ten cells, two per vessel cross-section. VEGF gradient extends in the direction of the sprout. Each cell indicated by a different color.

During angiogenesis, the topology of the network is shaped essentially through the dynamic process of stalk cells turning into a new tip cell (tip cell selection), which is dependent on the Delta-Notch lateral inhibition pathway, a widely used machinery to regulate aspects of development that require temporal control by a 'molecular clock'. For example, during somitogenesis, Delta-Notch oscillations determine the frequency of new somite formation ('the somite clock') (*Aulehla and Pour-quié, 2008*). In endothelial cells, we propose that modulation of any of the components of the central pattern generator will result in an altered pace of Delta-notch oscillations and an altered vascular patterning. Deceleration of the Delta-notch feedback loop (by a 'slow molecular metronome') will lead to the selection of fewer tip cells within a certain time frame and thus to the formation of a less dense network with bigger pore sizes. Acceleration of the selection process (by a 'fast molecular metronome') would result in an overly dense network (*Figure 4—figure supplement 3*). However, complex nonlinear feedback dynamics are often hard to intuit, and further careful simulation integrated with experimentation will be required to fully elucidate the temporal modulations and topological outcomes possible.

In this work, we describe how Sema3E-Plexin-D1 signaling can modify vascular density by impinging on the central pattern generator VEGF/Notch signaling. Our computational modeling predictions, mouse genetics mosaic analysis, and live imaging of individual cell dynamics in actively forming blood vessel networks and computational quantification of branching points show that the lack of Sema3E-Plexin-D1 signaling slows down the rate of tip cell selection and rearrangement, resulting in a less branched vascular network. Therefore, the Sema3E-Plexin-D1 pathway represents a 'faster molecular metronome' that results in a relatively dense network. These data suggest that *temporal* regulation of this critical, iterative aspect of network formation could be a general mechanism, and additional temporal regulators with varying pace (fast vs. slow) may exist to sculpt vascular topology in different tissues. Furthermore, our findings may provide insights into our understanding of morphogenesis in general, and aid in efforts to develop therapeutic approaches for tissue engineering and control of tumor progression and vascular diseases.

## Material and methods

### Animals

*Plxnd1*<sup>flox/flox</sup> mice (*Zhang et al., 2009*), *Plxnd1*<sup>+/-</sup> mice (*Gu et al., 2005*) and *Cdh5-Cre-ER*<sup>T2</sup> mice (*Monvoisin et al., 2006*) were maintained on a C57Bl/6 background. *Z/EG+* reporter (*Novak et al., 2000*) mice were maintained on a 129P3J;C57Bl/6 mixed background. Pregnant mice were obtained following overnight mating (day of vaginal plug was defined as embryonic day 0.5). All animals were treated according to institutional and US National Institutes of Health (NIH) guidelines approved by the Institutional Animal Care and Use Committee (IACUC) at Harvard Medical School.

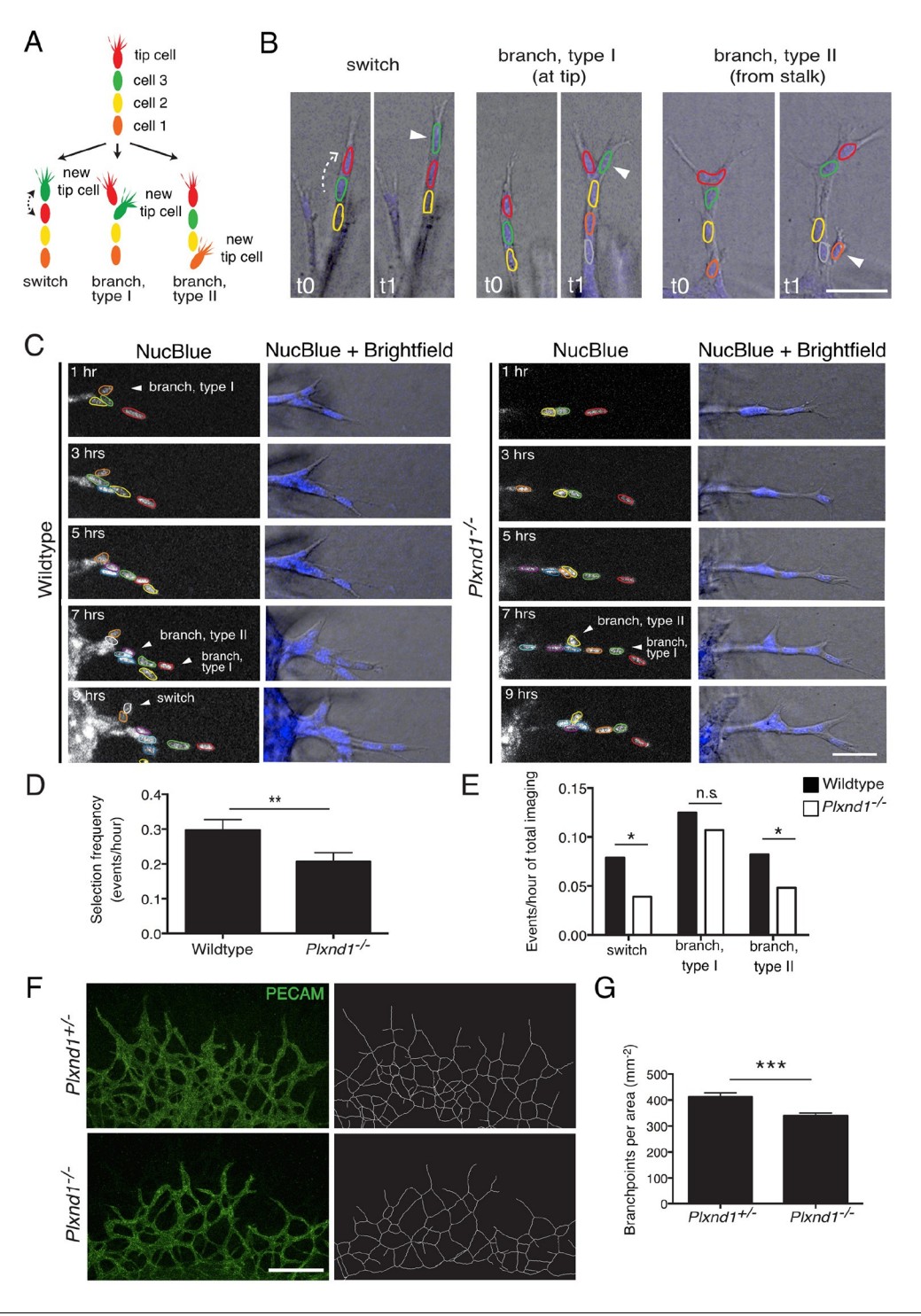

**Figure 4.** Endothelial cell live tracking in ex vivo lung explants reveals a reduction in tip cell selection frequency and a less branched lung vascular network in the absence of Sema3E-Plexin-D1 signaling. (**A**) Different types of tip cell selection events observed during live imaging. (**B**) Single frames from live imaging experiments illustrating the different types of tip cell selection events. Arrowheads point out the newly selected tip cells. Nuclei: blue. Scale bar: 50 μm. (**C**) Long-term live imaging experiments of vascular sprouts from wildtype and *Plxnd1⁻/⁻* lung explants. Single planes from z-stacks are shown. Arrowheads indicate a tip cell selection event. Nuclei: blue (top and middle), white (bottom). Individual nuclei are outlined by different colors in the middle panel. (**D**) Quantification of tip cell selection frequency calculated as events per hour. Tip cell selection frequency is reduced by factor 1.5 in

*Figure 4 continued on next page*

*Figure 4 continued*

sprouts from *Plxnd1⁻/⁻* explants. WT, n=24 sprouts from 12 explants. *Plxnd1⁻/⁻*, n=30 sprouts from 11 explants. Data is represented as mean +/- SEM. (**E**) Quantification of tip cell selection frequency calculated as incidence of events in each category as illustrated in (**A**) during total imaging time. **p=0.013 (**D**), *p=0.029 (e 'switch'), 0.041 (e 'branch, type II'), permutation test with shuffled genotypes. (**F**) Vascular sprouts originating from *Plxnd1⁺/⁻* and *Plxnd1⁻/⁻* lung explants on day 3. Left: whole-mount vascular staining (green, PECAM), right: reconstructed/ skeletonized network, Scale bar: 250 μm. (**G**) Quantification of branching points per area. The number of branching points is significantly reduced in *Plxnd1⁻/⁻* lung explants. Data is represented as mean +/- SEM. *Plxnd1⁺/⁻*, n=7 explants; *Plxnd1⁻/⁻*, n=6 explants. ***p=0.0005, permutation test with shuffled genotypes.

The following figure supplements are available for figure 4:

**Figure supplement 1.** The ex vivo sprouting assay.
**Figure supplement 2.** Computational method used for vascular network analysis.
**Figure supplement 3.** Model: Modifications of the central pattern generator lead to the formation of diverse vascular topologies.

## Mosaic retina analysis

Cre-mediated recombination was induced by intraperitoneal injection of tamoxifen (T5648, Sigma-Aldrich) dissolved in safflower oil on postnatal day (P) 4. Due to the different Cre sensitivities at *Plxnd1* and Z/EG locus, we experimentally determined the dosage of tamoxifen necessary for 45% of mosaicism. 25 pg of tamoxifen was injected in *Cdh5-Cre-ER^T2^; Plxnd1^flox/flox^* pups, and 10 ng of tamoxifen was injected in *Cdh5-Cre-ER^T2^; Z/EG+* pups. Mice were sacrificed at P5 and retinas were isolated for analysis. In situ hybridization, Isolectin B4 staining and ERG immunohistochemistry (1:200; SC353, Santa Cruz) were performed as described previously. Images of flat mounted retinas were taken at 40x magnification using Zeiss LSM 510 META confocal microscope. Images were processed using Adobe Photoshop and Image J (National Institutes of Health). Mosaic recombination in *Cdh5-Cre-ER^T2^;Plxnd1^flox/flox^* or *Z/EG; Cdh5-Cre-ER^T2^* was analyzed by in situ signal or GFP positivity, respectively, in combination with Isolectin B4 and endothelial nuclear ERG staining. The tip cells were determined as blind-ended endothelial cells that are associated with filopodia protrusions at the sprouting front. Endothelial cells (either tip or stalk cells) were counted by combination of Erg positive staining and morphological definition. 84 endothelial cells from 3 *Cdh5-Cre-ER^T2^;Plxnd1^flox/flox^* animals and 64 endothelial cells from 3 wildtype animals were counted for tip cells and stalk cells that were either *Plxnd1* positive or negative. 37 endothelial cells from 2 *Z/EG;Cdh5-Cre-ER^T2^* animals were counted for tip cells and stalk cells that were either GFP positive or negative. Statistical significance was tested using two-way Anova.

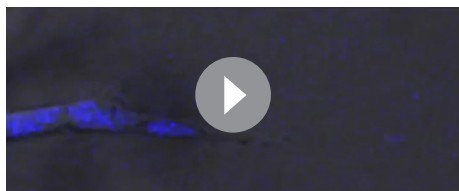

**Video 11.** Wildtype sprout in ex vivo endothelial cell tracking assay. The movie represents 9 hr of live imaging and corresponds to *Figure 4C*, left panel. Merge of fluorescent channel (nuclear live stain) and brightfield channel. Arrows indicate newly selected tip cells that give rise to a new sprout. Arrowhead indicates a switching event. Single planes from z-stacks are shown.

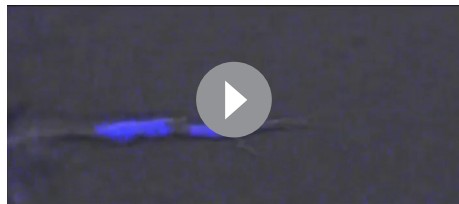

**Video 12.** *Plxnd1⁻/⁻* sprout in ex vivo endothelial cell tracking assay. The movie represents 9 hr of live imaging and corresponds to *Figure 4C*, right panel. Merge of fluorescent channel (nuclear live stain) and brightfield channel. Arrows indicate newly selected tip cells that give rise to a new sprout. Single planes from z-stacks are shown.

## RT-PCR

Total RNA was extracted from collagen embedded lung explants, or whole retina using the RNeasy Micro Plus RNA extraction kit (Qiagen) according to manufacturer's instructions. cDNA was generated from 100 ng total RNA using the Superscript III reverse transcription kit (Invitrogen). The following primers were used to detect *Plxnd1* and *Sema3e* transcripts: *Sema3e* forward 5'-aggctacgcctgtcacataaa-3', *Sema3e* reverse 5'-ccgttcttgatactcatccagc-3'; *Plxnd1* forward 5'-gctgactg-tagcctatgggga-3',*Plxnd1* reverse 5'- gccatctggtgggatgtcat-3'.

## Ex vivo endothelial cell tracking assay

Lungs were dissected from E12.0 embryos in ice-cold dissecting medium (DMEM + Penicillin/Strep-tomycin), divided into single lobes, washed in dissecting medium (30 min, 4°C) and embedded in a glass-bottom dish (Mattek) between two layers of polymerized collagen IA gel (1.5 mg/ml, Cellmatrix) prepared as previously described(*Jakobsson et al., 2006*). After solidification of collagen (30 min, 37°C), imaging medium was added (DMEM without phenol red, 15% ES cell grade FBS, 30 ng/ml rhVEGF (R&D systems), antibiotics). Explants were grown overnight at 37°C, 5% $CO_2$. Next day, nuclear live stain solution (NucBlue, Life Technologies, Carlsbad, CA) was added (2 drops/ml for 45 min). Imaging medium was replaced and explants were imaged immediately using a Leica SP8 confocal microscope equipped with a Tokai Hit chamber (37°C, 5% $CO_2$). 70 µm stacks were acquired at 20x magnification every 10 min. Single cells were tracked manually using a combination of nuclear staining and bright-field images of the sprouts. Per explant, 1–4 sprouts were analyzed for 9–20 hr. Tip cell selection frequency was calculated as selection events observed per hour live imaging. All 3 types of tip cell selection events were considered for quantification of selection frequency. Statistical significance was tested using a permutation test with shuffled genotypes.

## Immunostaining of lung explants

Explants were dissected at E12.0 or E13.5, prepared as described above, and fixed on day 2 or 3 in 4% PFA for 1 hr at RT. Then they were washed in PBS (3 x 10 min), permeabilized in blocking solution (PBS + 2% BSA + 0.2% Triton) for 1 hr at RT (2 x 30 min) and incubated with PECAM (1:300, BD), Plexin-D1 (abcam) or Dll4 (R&D) primary antibody in PBS + 0.2% Triton overnight at 4°C and additional 6 hrs at RT. Explants were then washed in PBS + 0.1% Triton (6x 30 min), incubated with secondary antibody in PBS + 0.1% Triton overnight at 4°C and washed in PBS + 0.1% Triton (2 x 30 min). If HRP-conjugated secondary antibody was used, explants were washed longer (3 x 30 min in PBS) and stained with DAB staining solution (1 mg/ml Diaminobenzindine tetrahydrochloride in PBS, Sigma) for 3 min. For network topology analysis, Alexa488 conjugated secondary antibody was used and explants were imaged at a Leica SP8 confocal microscope at 10x magnification.

## Computational modeling

In the previously established 'MemAgent-Spring Model' (MSM) each endothelial cell is comprised of multiple smaller computational elements ('agents') that represent local sections of cell membrane and actin tension beneath. The memAgents have dynamic internal levels of proteins, which enable them to sense protein levels in the local extracellular environment (primarily VEGF ligands). The endothelial cell integrates this local spatial information to determine its behavior (e.g. extension of filopodia) and perform genetic regulatory processes (e.g. Dll4-Notch) after a time delay representing the processes of transcription and translation (parameters D1 and D2 in *Figure 1B*). As the time delay for Notch/VEGF gene expression is not currently known in the mouse, the model was calibrated to match the known Notch periodicity (30 min) of the zebrafish somite clock (*Guidicelli and Lewis, 2004*) so D1=D2=28 time steps (representing 7 min of real time) as it was previously shown that, consistent with period/delay relations in the Notch somite clock models (*Guidicelli and Lewis, 2004*), the periodicity of Notch/VEGF signaling in the MSM model = 2 x (D1+D2+R1+R2), where R1 and R2 are the recovery delays, representing degradation rates, which were both set to 1 (*Bentley et. al., 2008*).

To include Sema3E-Plexin-D1 interactions to the existing MSM model of VEGF/Notch/Dll4 signaling in endothelial cells during tip/stalk selection (full model described in [*Bentley et al., 2008*; *2009*]), four new parameters were included. Two new time delay parameters: D3 controls how long it takes for VEGFR-2 to increase Plexin-D1 protein levels at the membrane and D4 determines the

time it takes for an active Plexin-D1 receptor to lower Dll4 expression. Additionally $s$ was added to determine the strength of Plexin-D1 down-regulation of Dll4 (specifically how many fewer Dll4 are produced for one active Plexin-D1 receptor) and $r3$, which controls how long the down regulation effect lasts for, encompassing the factors such as Plexin-D1 degradation rate, see *Figure 1B* for schematic.

As Sema3E is assumed to be uniformly present around the cell based on experimental data in the mouse retina (*Kim et al., 2011*), activation of Plexin-D1 by Sema3E is not directly modeled, but simply assumed to occur at a constant level. As the exact number of Plexin-D1 receptors on the cell surface is also not known we assume Plexin-D1 receptors vary within the same range as VEGFR-2 receptors (see (*Bentley et al., 2008*) for details), and are instantly activated by Sema3E when present. These assumptions produce the most parsimonious model possible, permitting Plexin-D1 levels to be controlled by just the D3 time delay parameter, and Sema3E-Plexin-D1 signaling strength to be determined by the modulation of a single parameter $s$, which varies the strength of effect of the signaling on Dll4 up-regulation specifically. Dll4 levels were then determined as follows:

$$Dll4_{t+1} = Dll4_t + V^{''}\delta - P^{''}s$$

where $V''$ is the number of active VEGFR-2 receptors by VEGF after time delay D1 has been applied, representing the current active VEGFR-2 level affecting gene expression in the nucleus. Likewise $P''$ is the number of Plexin-D1 receptors (assumed activated by Sema3E) able to affect gene expression after time delay D4.

### Calibrating Dll4 regulation parameters $\delta, s$

Previously δ, which represents the up-regulation strength of Dll4 by VEGF-VEGFR-2 signaling, was calibrated to 2 to generate matching tip/stalk pattern selection and sprouting behavior in vivo under different conditions (*Bentley et al., 2008*; *2009*). So now with a balancing inhibition term $s$ representing reduction in Dll4 via Plexin-D1, we know that δ - s = 2 is required for normal sprouting. Any combination of δ and $s$ values such that this relation held true would give normal sprouting. To simulate loss of Sema3E-Plexin-D1 signaling $s$ was set to zero. Thus the value for δ chosen ultimately determines the strength of the simulated Sema3E-Plexin-D1 mutant phenotype, hence results are shown throughout across a range of δ values when s = 0. For control simulations s = δ-2.

### Calibrating time delay parameters D3 and D4

Experiments indicate that the rate of Plexin-D1 up-regulation by pVEGFR-2 is fast compared to pVEGFR-2 up-regulation of Dll4, indicating that together the delays D3+D4 >=D1. To investigate the effects of varying the temporal regulation of Plexin-D1 on tip cell selection a sensitivity analysis was performed simulating with different delay settings for the new parameters D3 and D4 (a full analysis of varying delays D1 and D2 is given in [*Bentley et al., 2008*]). It was found that the lateral inhibition mechanism is strictly sensitive to the values of these new delay parameters relative to the existing delays D1 and D1. In the model, only a setting of D3+D4 =D1 would allow for normal tip cell selection in control conditions (*Video 1*). Even a delay with D3 or D4 = ± 1 timestep in the model (representing 15 s) would disrupt the process and tip cells could not be selected and the system falls into unrealistic 'flashing' oscillations as the cells instantly raise and then lower dll4 each time step through the Notch/VEGF negative feedback loop resulting in a counter intuitive hypersprouting rather than inhibited phenotype as no cell is under the inhibition long enough to become a stalk cell (*Video 2*, *3*). Interestingly if all delay parameters are set to zero, representing the null hypothesis that no time delays are required to explain the phenotype, the same system behavior occurs as in *Video 2* and *3* illustrating the importance of explicitly representing the amount of time that gene expression takes in computational models.

### Calibrating the degradation rate *r3*

Disrupting the degradation rate *r3* of Plexin-D1 was also found to have drastic affects on the ability of the system to select tip cells. The *r3* parameter was required to satisfy: *r3 = r2 = r1 = 1* timestep (representing 15 s). Any increase led to similar irregular flashing oscillations and abrogated tip cell selection as seen with increases to the delays D3 or D4 (*Video 4* and *5*). Thus for all simulations D3 +D4 =D1, where D3= 1 and D4 = 27 timesteps.

## Mosaic vessel simulations

Mosaic vessels follow the same simulation method as simulations of a fully wildtype or mutant vessel, except that at the start of the simulation each cell is randomly assigned a wildtype or mutant setting of the δ and s parameters (*Video 8*). 45% mosaicism was calculated as average of simulations with 40% and 50% mosaicism. Results were averaged over 50 runs.

## The 'MSM-CPM' model

In this model a cell can move within the adhered collective of the sprout powered by multiple local junctional adhesion movements (based on the Cellular Potts Model 'CPM' of differential adhesion [*Graner and Glazier, 1992*]), which are regulated by VEGF/Notch signaling.

## Computational method used for vascular network analysis

The stacks were processed and analyzed in 3D using the following Python 2.7 modules: Numpy, Scipy, Matplotlib, Opencv2, Igraph and Networkx.

The first step of the algorithm was to apply a Gaussian smoothing filter to the stacks. A standard deviation of 5 μm was used for the Gaussian kernel. Next, the moment-preserving threshold technique (*Tsai, 1985*) was used in order to find a proper threshold for stack binarization. Pixels having intensity values larger than the calculated threshold were classified as belonging to a vessel. Remaining image components(*Stockman, and Shapiro, 2001*) smaller than 20000 $\mu m^3$ were considered background noise and removed from the binary image. A thinning procedure (*Palagyi and Kuba, 1998*) was then applied to the binary image, resulting in what we call the skeleton (*Costa and Cesar, 2009*) of the blood vessels. The skeleton tends to present some sets of connected pixels having more than two neighbors each. Such sets were erased from the skeleton and represented as a single pixel at the center of mass of the set. The remaining skeleton pixels having at least three neighbors were classified as branching points, while pixels having one neighbor were considered a termination point.

Spurious skeleton segments were removed by an iterative algorithm. First, termination segments smaller than 20 μm were erased, where a termination segment is defined as a segment having one termination point. After erasing such segments, new small termination segments might appear. They were iteratively erased until no new termination segments smaller than 20 μm remained. The sample was then characterized by quantification of the remaining branching points. Finally, in order to validate the analysis, we created images containing both the original image and the final skeletons and verified that the obtained skeletons were accurately representing the original blood vessel structure. Statistical significance was tested using a permutation test with shuffled genotypes.

## Acknowledgements

We thank Drs. Christopher Harvey, Jonathan Cohen, Ayal Ben-Zvi, and members of the Gu laboratory for comments on the manuscript. We thank Dr. Christopher Harvey for statistical analysis, and Rob Rosa for technical help on lung explant imaging. We thank S Demaki for providing the *Z/EG+* reporter mice, Tom Jessell for providing *Plxnd1^flox/flox^* mice, Luisa Iruela-Arispe for providing the *Cdh5-Cre-ER^T2^* mice. The Enhanced Neuroimaging Core at Harvard Neurodiscovery Center for help with imaging and advice on image analysis. This work was supported by: DFG-German Research Foundation postdoctoral fellowship KU 3081/1&2 (EK); NRSA Postdoctoral training grant T32 NS07484-12 (JK); Brooks postdoctoral fellowship (AT); FAPESP grants 11/22639-8 (CHC) and 11/50761-2 (LdaFC) and CNPq grant 307333/2013-2 (LdaFC); BIDMC and the Leducq transatlantic Network ARTEMIS (KB); and the flowing grants to CG: NIH grants R01NS064583, DP1 NS092473, the BrightFocus Foundation, and a Kaneb Fellowship.

## Additional information

### Funding

| Funder | Grant reference number | Author |
|---|---|---|
| Deutsche Forschungsgemeinschaft | KU 3081/1-1 | Esther Kur |
| National Institutes of Health | R01NS064583 | Chenghua Gu |
| BrightFocus Foundation | | Chenghua Gu |
| National Institutes of Health | DP1 NS092473 | Chenghua Gu |

The funders had no role in study design, data collection and interpretation, or the decision to submit the work for publication.

### Author contributions

EK, JK, Conception and design, Acquisition of data, Analysis and interpretation of data, Drafting or revising the article; AT, CHC, KIH, LdFC, Acquisition of data, Analysis and interpretation of data, Drafting or revising the article; KB, CG, Conception and design, Analysis and interpretation of data, Drafting or revising the article

### Author ORCIDs

Chenghua Gu, http://orcid.org/0000-0002-4212-7232

### Ethics

Animal experimentation: All animals were treated according to institutional and US National Institutes of Health (NIH) guidelines approved by the Institutional Animal Care and Use Committee (IACUC) protocols (# 04146) at Harvard Medical School.

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
