## [Decision Letter]

Thank you for submitting your manuscript "Temporal modulation of collective cell behavior controls vascular network topology" as a Short Report to *eLife*. Three experts reviewed your manuscript, and their assessments, together with my own (Jeremy Nathans, Reviewing Editor), form the basis of this letter. The process was overseen by Janet Rossant as Senior Editor.

As you will see, all of the reviewers were impressed with the importance and novelty of your work. We look forward to receiving the revised version of your manuscript.

I am including the three reviews (lightly edited) at the end of this letter, as there are several specific and useful suggestions in them.

Reviewer #1:

In this manuscript, Kur et al. describe the use of an improved computational modeling in combination with ex vivo time-lapse microscopy to identify the role of Sema3E/Plexin-D1 in the dynamic selection of tip cells in the developing vasculature. According to the suggested model, inhibition of Dll4 expression due to Plexin-D1 activation by Sema3E result in reduced lateral inhibition of tip cell phenotype in neighboring stalk cells. This, in turn, results in more tip cell selection and more complex vasculature development. To support the model, the authors use two experimental systems. First, they observed that in a *Plxnd1* mosaic vasculature, *Plxnd1^-/-^* endothelial cells are significantly more likely to assume tip cell phenotype, as the model predicts for the case of reduced Dll4 inhibition. Second, they use lung explant to monitor the dynamics of endothelial sprout development in the presence or absence of Plexin-D1 signaling. As the model predicts, in the absence of Plexin-D1 signaling (in *Plxnd1^-/-^* lungs) the authors observe reduced rate of tip cell selection that result in less elaborate endothelial network development.

This is a very interesting manuscript and it is well written. It provides an excellent description of the dynamic of endothelial tip cell selection.

Below are some notes that may help improve the manuscript.

1) The model depicted in Figure 1 is lacking at least one major component. The underdeveloped vasculature observed in the *Sema3e^-/-^* or the *Plxnd1^-/-^* mouse retina will result in local hypoxia and thus increase in VEGF production by the surrounding tissue. This is likely to increase Dll4 expression in "stalk" cells and inhibition of tip cell selection irrespective of the status of Plexin-D1 activation.

2) The authors make excellent use of the time-lapse microscopy of lung explants to monitor the dynamics of tip cell selections. The cultures, however, could be better characterized. First, IHC for Plexn-D1 should be performed to show that the sprouting endothelial cells express the receptor. Second, IHC for Dll4 should be performed to show potential differential expression of Dll4 by tip cells, as the model predicts. Third, it is not clear how Sema3E signaling functions in the explant system. Supposedly the tissue surrounding the endothelial cells expresses Sema3E but these cells sprout to a considerable distance from the explant. Is there enough Sema3E secreted to the medium for signaling? Could manipulation of the concentration of Sema3E (either addition of Sema3E to the medium or use of neutralizing antibodies) affect the sprouting endothelial cell behavior? Forth, it is worth mentioning that in the explant, in contrast to the in vivo situation, there is, most likely, no gradient of VEGF signaling since the media is supplemented with VEGF.

3) A minor point: in the text the authors state that the lung explants were cultured in VEGF-conditioned medium while according to the Materials and methods section recombinant human VEGF was added to the medium.

4) Figure 2, replace chimera with mosaic. These terms are, not interchangeable. A chimera is made of cells from two different individuals.

Reviewer #2:

This is a well written, interesting paper about the regulation of vascular network density, an important yet poorly understood process. In this collaborative work between the Gu and Bentley labs, each offering unique expertise in experimental and computational modeling studies, the authors report that temporal regulation of the VEGF/Notch feedback loop is key to determining vascular density. Namely, they find that the rate of tip-cell selection dictates network density. More specifically, they manipulate the Sema3E-Plexin-D1 signaling pathway, which had been previously shown by the Gu lab to regulate developmental angiogenesis, to support their claims.

[Please note that while I can appreciate the value of the computational studies, I cannot evaluate them critically.]

The experimental work appears to be sound and Figure 2 (cell autonomy studies) and 4 (live tracking of endothelial cell behavior) represent new and informative data.

The question of whether this paper represents a sufficiently novel concept/set of observations to warrant publication in *eLife* will be more easily addressed once the novelty and quality of the computational studies are critically evaluated.

Reviewer #3:

To me this is an outstanding manuscript because it is the first time where I have seen computational modeling making a contribution beyond confirming already known experimental evidence. By introducing a signaling time delay the authors have added an additional level of sophistication to their computational model that allows them to explore a mechanism in vitro that is difficult to approach experimentally. Based on this the authors suggest that reduced speed of signaling can explain the vascular phenotype in Sema3E loss of function mice. The manuscript is of exceptional high quality, genuinely original and a joy to read.

My main criticism is that the "speed of signaling hypothesis" has not been properly compared to alternative hypotheses. The authors have previously shown that Sema3E has an inhibitory effect on Dll4-Notch signaling. Here they tested how the temporal aspect of that signaling affects vessel development. But it could be argued that the simple inhibitory effect (S) of Plexin-D1 signaling could have a similar effect. So, I wonder whether the outcome of in silico deletion of Plexin-D1 has the same outcome when the D levels are reduced and S is increased (i.e. is the temporal aspect of signaling in the model really needed?). In other words, the authors argue that by taking into account the speed of signaling their model has been improved, but they should compare the new with the old model and show why the new one is better and where the old one falls down.

---

## [Author Response]

Reviewer #1:

Below are some notes that may help improve the manuscript.1) The model depicted in Figure 1 is lacking at least one major component. The underdeveloped vasculature observed in the Sema3e^-/-^or the Plxnd1^-/-^

*mouse retina will result in local hypoxia and thus increase in VEGF production by the surrounding tissue. This is likely to increase Dll4 expression in "stalk" cells and inhibition of tip cell selection irrespective of the status of Plexin-D1 activation.*

This is a very interesting suggestion. If we step through and consider it logically however we arrive at the current depiction of the figure being sufficient, though we agree it worth considering carefully. Firstly, the phenotype of a retina with high VEGF is known and previously published (e.g. by ourselves in Bentley K et al. Nature Cell biology 2014). A key element of the high VEGF retinal phenotype is the switch from branching to drastic expansion and widening of the leading edge, which mimics the expansion of abnormal vessels in pathologies with high VEGF, furthermore large patches of over-activated cells can be observed in certain regions (also made clear in Bentley et al. 2008). Therefore, given the phenotype here exhibits no such expansion, no over activated regions and branching, though reduced, continues as normal when initiated we can consider that even if there is an elevation in VEGF it is not sufficiently high to overcome the reduction in activation triggered by Semaphorin/Plexin-D1 loss and so not useful to include in a parsimonious model of the primary factors affecting this phenotype.

2) The authors make excellent use of the time-lapse microscopy of lung explants to monitor the dynamics of tip cell selections. The cultures, however, could be better characterized. First, IHC for Plexn-D1 should be performed to show that the sprouting endothelial cells express the receptor. Second, IHC for Dll4 should be performed to show potential differential expression of Dll4 by tip cells, as the model predicts. Third, it is not clear how Sema3E signaling functions in the explant system. Supposedly the tissue surrounding the endothelial cells expresses Sema3E but these cells sprout to a considerable distance from the explant. Is there enough Sema3E secreted to the medium for signaling? Could manipulation of the concentration of Sema3E (either addition of Sema3E to the medium or use of neutralizing antibodies) affect the sprouting endothelial cell behavior? Forth, it is worth mentioning that in the explant, in contrast to the in vivo situation, there is, most likely, no gradient of VEGF signaling since the media is supplemented with VEGF.

A) We performed double immunohistochemistry staining with Plexin-D1 and PECAM antibodies on lung explants and found Plexin-D1 protein in sprouting endothelial cells. We added this result as a new panel in Figure 4—figure supplement 1) and added figure legend, also modified the eighth paragraph of the main text.

B) We performed double immunohistochemistry with Dll4 and PECAM antibodies on lung explants. We observed that in Wildtype endothelial sprouts Dll4 level is much higher in the tip cells compared to stalk cells. In the *Plxnd1^-/-^*sprouts we observed more high-Dll4-expressing cells compared to wildtype. These data are shown in Figure 4—figure supplement 1 and added figure legend, also modified the eighth paragraph of the main text.

C) Since there is no good Sema3E antibody available, it is hard to measure the protein levels in the explants. The reviewer’s suggestion of doing live imaging analysis under different concentrations of Sema3E is an excellent one, and we just begin these experiments as part of a future study to dig deeper into the molecular details of Sema3E-Plexin-D1 signaling in this context. Our preliminary data showed that adding Sema3E to the medium resulted in an increased frequency of speed changes. However, to perform these live imaging experiments rigorously with different concentrations of Sema3E and with enough sample size will take a long time, which will be part of the future follow up study.

D). Yes, there is no gradient of VEGF in the medium. We have added a sentence about this in the eighth paragraph of the main text.

3) A minor point: in the text the authors state that the lung explants were cultured in VEGF-conditioned medium while according to the Materials and methods section recombinant human VEGF was added to the medium.

We thank the reviewer for the comment. Indeed, recombinant human VEGF was added to the medium and we corrected text accordingly (main text, eighth paragraph).

4) Figure 2, replace chimera with mosaic. These terms are, not interchangeable. A chimera is made of cells from two different individuals.

Response: We thank the reviewer for this comment. We have made the change in Figure 2 according to the reviewer’s suggestion.

Reviewer #3:My main criticism is that the "speed of signaling hypothesis" has not been properly compared to alternative hypotheses. The authors have previously shown that Sema3E has an inhibitory effect on Dll4-Notch signaling. Here they tested how the temporal aspect of that signaling affects vessel development. But it could be argued that the simple inhibitory effect (S) of Plexin-D1 signaling could have a similar effect. So, I wonder whether the outcome of in silico deletion of Plexin-D1 has the same outcome when the D levels are reduced and S is increased (i.e. is the temporal aspect of signaling in the model really needed?). In other words, the authors argue that by taking into account the speed of signaling their model has been improved, but they should compare the new with the old model and show why the new one is better and where the old one falls down.

We sincerely thank the reviewer for their supportive comments on the modeling.

To clarify, the original model, as published in Bentley, K. et al. J. Theor. Biol (2008) and Bentley, K. et al. Plos Comp. Biol (2009) and in all subsequent papers, there was actually always a time delay component to gene expression, this isn’t a new feature of this study, however re-reading our modeling introduction we understand why it may have come across that it was. Time delays were present in the original model to capture the regulation that VEGF activation has on Dll4 (delay D1 in the original model) and Notch activation has on VEGFR-2 (delay D2) as gene expression takes on the on the order of half an hour to multiple hours and the model time steps represent 15 seconds these need to be included to be true to the biological case. Thus when extending the model here to include Semaphorin-Plexin signaling it was necessary to also include the corresponding time delay parameters (the new ones being D3 and D4) but time delay wasn’t a novel feature of this study per se, rather the inclusion of Semaphorin effects on Dll4 was the novelty. Firstly, to help clarify this in the main text we have adjusted the third paragraph. Likewise, we have adjusted the Methods text accordingly.

However, we agree a lot of models do side step the reality of the time delays and simulate gene expression effects as instantaneously felt at the cell surface, and anyway from a theoretical point of view it is certainly interesting to just see what happens when we do remove the time delays, to explore the null hypothesis that the reviewer suggests. So, even though it must be stressed such an assumption could never be a realistic alternative to the proposed model as it is not biologically plausible for gene expression to happen in an instant, we re-simulated the model with gene expression changes occurring instantaneously (removing D1-D4). As suggested by the reviewer, S needs to then be increased to compensate for the accumulation of Dll4 effects over time when time delays are present. The result is that the system behavior looks almost exactly like that in Video 3, where the time delays were instead mismatched. In a KO simulation without time delays (where δ is high to account for the high S) the cells flash rapidly between high and then low dll4 levels, as big changes in gene expression occur instantaneously every time step, the cells swing rapidly from strong inhibition to release that inhibition and back (when dll4 is high VEGFR-2 expression drops instantaneously at the next timestep driving an instant drop in Dll4 production at the next timestep which breaks the inhibition and allows the VEGFR-2 receptor levels to rise instantaneously) The negative feedback essentially causes an oscillation with period 1 timestep. The phenotypic result is interesting, as can be seen in Video 3, no cell is then continually inhibited so every other timestep extension of filopodia can continue, so the phenotype is counter intuitively one of hypersprouting, rather than inhibition. Therefore, we can reject the null hypothesis that a model without time delays in gene expression could still account for the Plexin KO phenotype, which of course does not hypersprout or hyposprout. To clarify this point the text “Calibrating time delay parameters D3 and D4”has been added to the Methods section on calibrating the time delays.